# Evolution of fast-growing piscivorous herring in the young Baltic Sea

Jake Goodall [1], Mats E. Pettersson [1], Ulf Bergström [2], Arianna Cocco[1], Bo Delling [3], Yvette Heimbrand [2], O. Magnus Karlsson [4], Josefine Larsson[5], Hannes Waldetoft [4], Andreas Wallberg [1], Lovisa Wennerström[2] & Leif Andersson [1,6] ✉

The circumstances under which species diversify to genetically distinct lineages is a fundamental question in biology. Atlantic herring (*Clupea harengus*) is an extremely abundant zooplanktivorous species that is subdivided into multiple ecotypes that differ regarding spawning time and genetic adaption to local environmental conditions such as temperature, salinity, and light conditions. Here we show using whole genome analysis that multiple populations of piscivorous (fish-eating) herring have evolved sympatrically after the colonization of the brackish Baltic Sea within the last 8000 years postglaciation. The piscivorous ecotype grows faster, and is much larger and less abundant than the zooplanktivorous Baltic herring. Lesions of the gill rakers in the piscivorous ecotype indicated incomplete adaptation to a fish diet. This niche expansion of herring in the young Baltic Sea, with its paucity of piscivorous species, suggests that empty niche space is more important than geographic isolation for the evolution of biodiversity.

An important contribution to the evolution of biodiversity occurs when a species forms an adaptive radiation after colonizing a new environment, for instance, an island with a paucity of species. A classic example is the evolution of 18 different species of Darwin's finches in the Galápagos archipelago from a single ancestral species within the last 1–2 million years[1,2]. Characteristic features of this evolutionary process include niche expansion, where subpopulations evolve by genetic adaptation to alternative food resources, which has occurred repeatedly among Darwin's finches[1]. Another important feature is character displacement, phenotypic changes that reduce competition between genetically distinct populations[3,4]. Similar adaptive radiations occur frequently in several species of salmonids after postglacial colonization of freshwater lakes[5]. Species from five different salmonid genera (*Coregonus, Oncorhynchus, Prosopium, Salmo,* and *Salvelinus*) tend to form genetically differentiated sympatric morphs or ecotypes showing niche expansion as well as character displacement[5]. For instance, there are four distinct ecotypes of Arctic charr in Lake Thingvallavatn in Iceland: large and small benthic morphs, as well as planktivorous and piscivorous pelagic morphs. An important question in evolutionary biology is whether such adaptive radiations may occur in sympatry or if some degree of allopatry is required. It is likely that the Galápagos geography–being an archipelago with many small islands where populations may become reproductively isolated for a period of time–has facilitated the speciation process of Darwin's finches. Similarly, it is possible that microallopatry may occur in lake systems where different morphs of fish evolve during at least partial reproductive isolation.

Atlantic herring (*Clupea harengus*) is an extremely abundant species with an estimated census population of $10^{12}$ (ref. 6). It is a zooplankton feeder with a key role in the North Atlantic ecosystem as a link between plankton production and higher trophic levels (piscivorous fish, sea birds, and marine mammals). Atlantic herring is a

[1]Department of Medical Biochemistry and Microbiology, Uppsala University, Uppsala, Sweden. [2]Department of Aquatic Resources, Swedish University of Agricultural Sciences, Uppsala, Sweden. [3]Department of Zoology, Swedish Museum of Natural History, Stockholm, Sweden. [4]IVL Swedish Environmental Research Institute, Stockholm, Sweden. [5]Marint centrum, Simrishamns kommun, Simrishamn, Sweden. [6]Department of Veterinary Integrative Biosciences, Texas A&M University, College Station, TX, USA. ✉e-mail: Leif.Andersson@imbim.uu.se

broadcast spawner, where large schools of fish aggregate in coastal areas to spawn. Females and males release eggs and sperm into the water column before the fertilized eggs stick to vegetation or seafloor, where development occurs under local environmental conditions. Initial genetic studies in the 1980s suggested that all herring in the North Atlantic and the brackish Baltic Sea could be a single panmictic population because no genetic differentiation was noted for 13 polymorphic loci[7]. This was a surprising finding, given Linnaeus classified the Baltic herring as *C. harengus membras* in distinction of the Atlantic herring (*C. harengus harengus*) based on morphological differences[8]. The Baltic herring is smaller, with less fat than the Atlantic herring, most likely due to less productive plankton production and possibly physiological stress due to the low salinity in the Baltic Sea[9,10]. This young sea has only existed for ~8000 years since the end of the last glaciation[11]. Recent studies based on whole-genome sequencing (WGS) confirmed a lack of genetic differentiation at selectively neutral markers but hundreds of loci underlying ecological adaptation related to variation in salinity, temperature, light conditions, and spawning time show genetic differentiation among ecotypes of Atlantic herring[12–14]. The $F_{ST}$ distribution, measuring the relative proportion of genetic variation within and between populations, deviates significantly from the one expected for selectively neutral alleles under a genetic drift model[15], implying an important role for natural selection in determining allele frequencies. Thus, Atlantic herring is subdivided into several genetically differentiated ecotypes that show differences in spawning locations, spawning time, migration patterns, and adaptation to local environmental conditions such as salinity and temperature[14].

This study aimed to explore genetic differentiation in a restricted area of the Baltic Sea (Fig. 1a), and in particular, to explore whether the herring denoted 'Slåttersill' by local communities constitutes a genetically unique population (Fig. 1b). Slåttersill is substantially larger than the dominating type of Baltic herring, where 'sill' refers to the Swedish name for the much larger Atlantic herring. This large type of herring has long been known to enter and spawn in coastal areas around hay harvest (Slåtter in Swedish) time in June. The presence of large herring in the Baltic Sea has also been noted by fishery biologists[16,17], but it is unknown if such large herring represent the tail of the size distribution of Baltic herring or constitute a distinct subpopulation. Here, we show that Slåttersill is a genetically unique ecotype of herring with altered feeding behavior and faster growth rate and that multiple subpopulations of such large, fish-eating herring exist in the Baltic Sea. This represents a remarkable example of sympatric differentiation and niche expansion in an extremely abundant broadcast spawner after colonizing a new environment within the last 8000 years.

## Results

### Slåttersill is a genetically distinct ecotype of Baltic herring

We collected multiple samples of both spring- and autumn-spawning planktivorous Baltic herring, as well as Slåttersill, at spawning from the same geographic region of the Baltic Sea (Fig. 1a, b, Supplementary Table 1). We used otoliths to determine the age of a subset of individuals from the three groups, which allowed us to establish growth curves (Fig. 1c). The data showed that the small spring- and autumn-spawning herring have similar growth curves, whereas Slåttersill has a faster growth rate, implying a different feeding behavior. Slåttersill were also significantly younger at capture (compared with spring-spawning Baltic herring: $\bar{x}-\bar{y}=3.4$ years, C.I. = 2.1–4.8 years, two-sided $t$ test, $t(49)=5.1$, $P=6.4\times10^{-8}$; compared to autumn-spawning Baltic herring: $\bar{x}-\bar{y}=2.5$ years, C.I. = 1.3–3.6 years, two-sided $t$ test, $t(49)=4.4$, $P=6.0\times10^{-8}$; Fig. 1c, Supplementary Table 2), demonstrating that Slåttersill are not simply older individuals at the tail-end of the small Baltic herring size distribution.

We genotyped all fish using a multi-species SNP chip that includes ~4500 SNPs for Atlantic herring, including most regions of the genome showing genetic differentiation among ecotypes[18]. A principal component analysis (PCA) showed two major clusters representing genetic differentiation between spring- and autumn-spawning herring across PC1 (Fig. 1d). Slåttersill, which spawns in mid-June, cluster with the spring-spawners but with a clear shift across PC2 revealing genetic differentiation. Thus, Slåttersill is a genetically distinct population, and the two replicates of this ecotype from 2022 and 2023 fall very closely in the PCA plot.

We next made a genome-wide, SNP-by-SNP contrast between Slåttersill and spring-spawning Baltic herring (Fig. 1e). This revealed four regions with highly significant differentiation ($P<10^{-10}$ after Bonferroni correction, per SNP $\chi^2$ test (d.f. = 1); Supplementary Fig. 1). The region on chromosome 17 overlaps one of four major inversions affecting ecological adaptation in Atlantic herring[14]. The regions on chromosomes 15 and 19 include several loci previously associated with variation in spawning timing[14]. In addition to these four major loci, a handful of additional regions reached genome-wide significance, corrected $P<0.05$ from per SNP $\chi^2$ test (d.f. = 1). A list of all loci showing significant differentiation between Slåttersill and spring-spawning Baltic herring is in Supplementary Data 1.

### Is Slåttersill a fish-eating (piscivorous) ecotype?

The discovery that Slåttersill is a genetically distinct ecotype of Baltic herring prompted us to perform a detailed phenotypic comparison of Slåttersill, spring- and autumn-spawning herring, and Atlantic herring (Supplementary Data 2, Supplementary Fig. 2). We confirmed a previously reported difference[19] that the number of vertebrae in the Baltic herring is smaller than that in the Atlantic herring ($\bar{x}-\bar{y}=-1.5$, C.I. = −1.9–1.0, two-sided $t$ test, $t(42)=-6.8$, $P=2.9\times10^{-8}$), and the Slåttersill conformed with the Baltic herring by having the lowest average number (Supplementary Data 2). The number of gill rakers was higher in Baltic herring than in our samples of Atlantic herring, and also here, Slåttersill were more similar to Baltic herring than to Atlantic herring ($\bar{x}-\bar{y}=5.7$, C.I. = 4.0–7.4, two-sided $t$ test, $t(38)=6.8$, $P=4.8\times10^{-8}$; Supplementary Data 2). The result is consistent with a previous study reporting higher gill raker counts in Baltic herring compared with most Atlantic herring populations[20]. Results from PCA of the morphometric data are given in Supplementary Table 3 and Supplementary Fig. 3. PC I mainly reflects the overall size and explains 93% of the variation in the sample. PC II and the following components reflect variation less correlated to the size. Along the PC II axes, Slåttersill is most similar to the Baltic spring-spawning herring. Studies of individual measurement reveal that several characteristics, e.g., body depth, caudal peduncle depth, head length, cranial length, and eye diameter, contribute to separation along the PC II axes. A larger head in Baltic herring is probably in part the result of fewer vertebrae giving a proportionally shorter body. Eye diameter is affected by growth rate[21] and comparing specimens of similar size, Baltic herring (spring and autumn) have larger eyes than their Atlantic counterpart, possibly indicating the slower growth rate in the former[22]. For the large-sized Slåttersill, compared to Atlantic herrings of similar size, there is no distinct difference, possibly as a result of faster growth in Slåttersill (Fig. 1c), resulting in proportionally smaller eyes.

A striking difference was that 8 out of 10 Slåttersill showed gill rakers with clear lesions (Fig. 2a, b). The morphology of gill rakers in planktivorous (plankton-feeding) and piscivorous fish is drastically different because the former have numerous dense gill rakers for sieving plankton. Among clupeiforms, comparing *C. harengus* to the

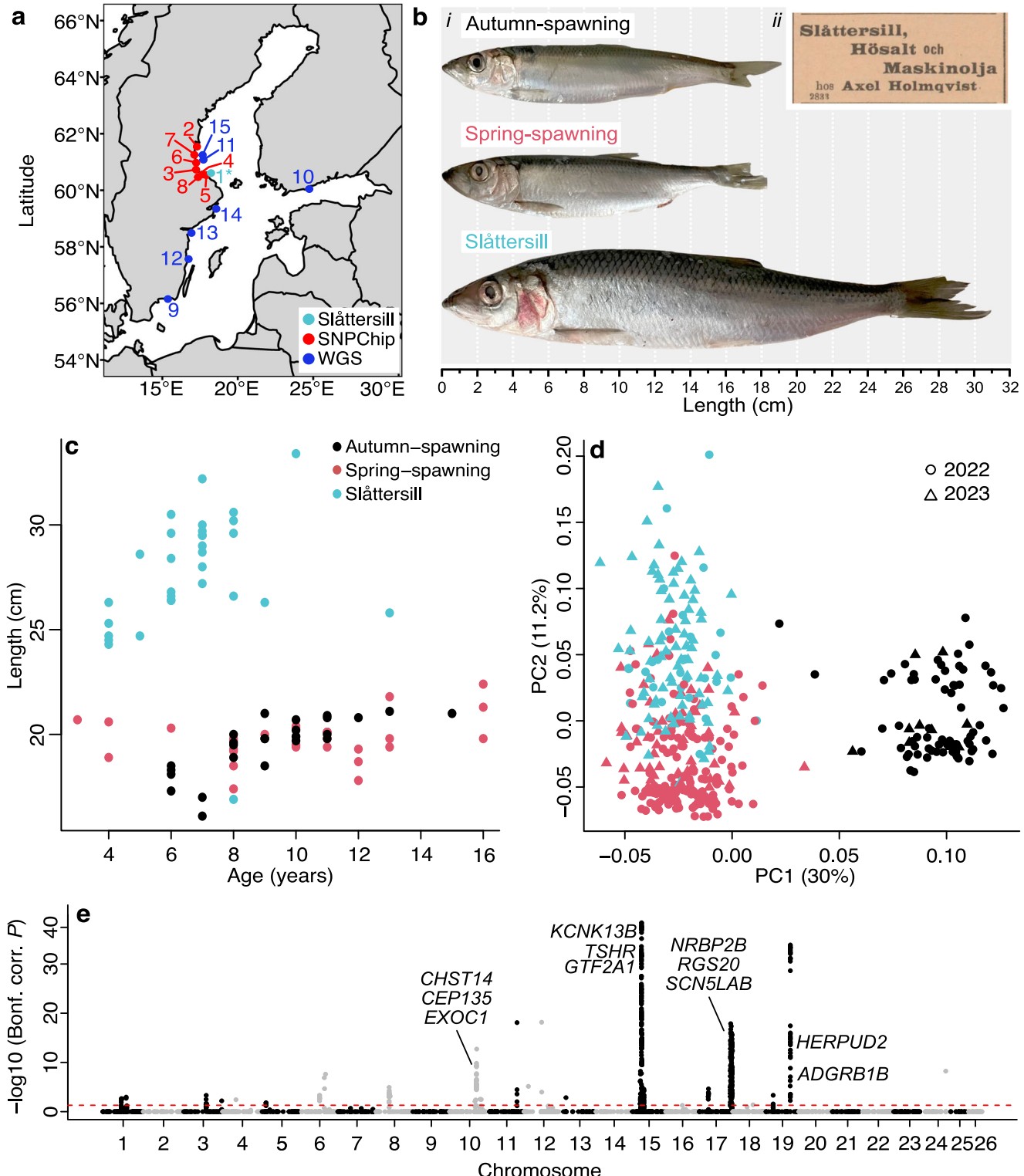

**Fig. 1 | Characterization of the large Slåttersill and reference populations of small spring-spawning and autumn-spawning Baltic herring. a** Map showing sampling locations in the Baltic Sea. Samples used for SNP-chip and whole-genome sequencing are given in red and blue numbers, respectively. Sample 1* (light blue) is Slåttersill used for both types of analysis. Explanations for localities are in Supplementary Tables 1 and 6. The underlying map was sourced from the public domain maps hosted by rnaturalearth (https://github.com/ropensci/rnaturalearth). **b** (i) Striking size differences between large Slåttersill and small spring-spawning and autumn-spawning Baltic herring. (ii) A newspaper clip showing an advertisement for 'Slåttersill' in the local newspaper Hudiksvallsposten from 1902-08-16. **c** Distribution of age and length among the three populations demonstrating differences in growth rate. **d** PCA plot based on genotype data for $n$ = 3840 SNPs. Colors as in **c. e** Genome-wide, SNP-by-SNP contrast between Slåttersill ($n$ = 107) and small spring-spawning Baltic herring from the Bothnian Sea ($n$ = 201) utilizing 3995 SNPs. $x$ axis: SNPs from individual chromosomes represented in alternating colors, $y$ axis: Bonferroni corrected $P$ values (−log10 transformed values from per SNP $\chi^2$ test (d.f. = 1), red dotted line equals corrected $P$ = 0.05). Source data are provided as a Source Data file.

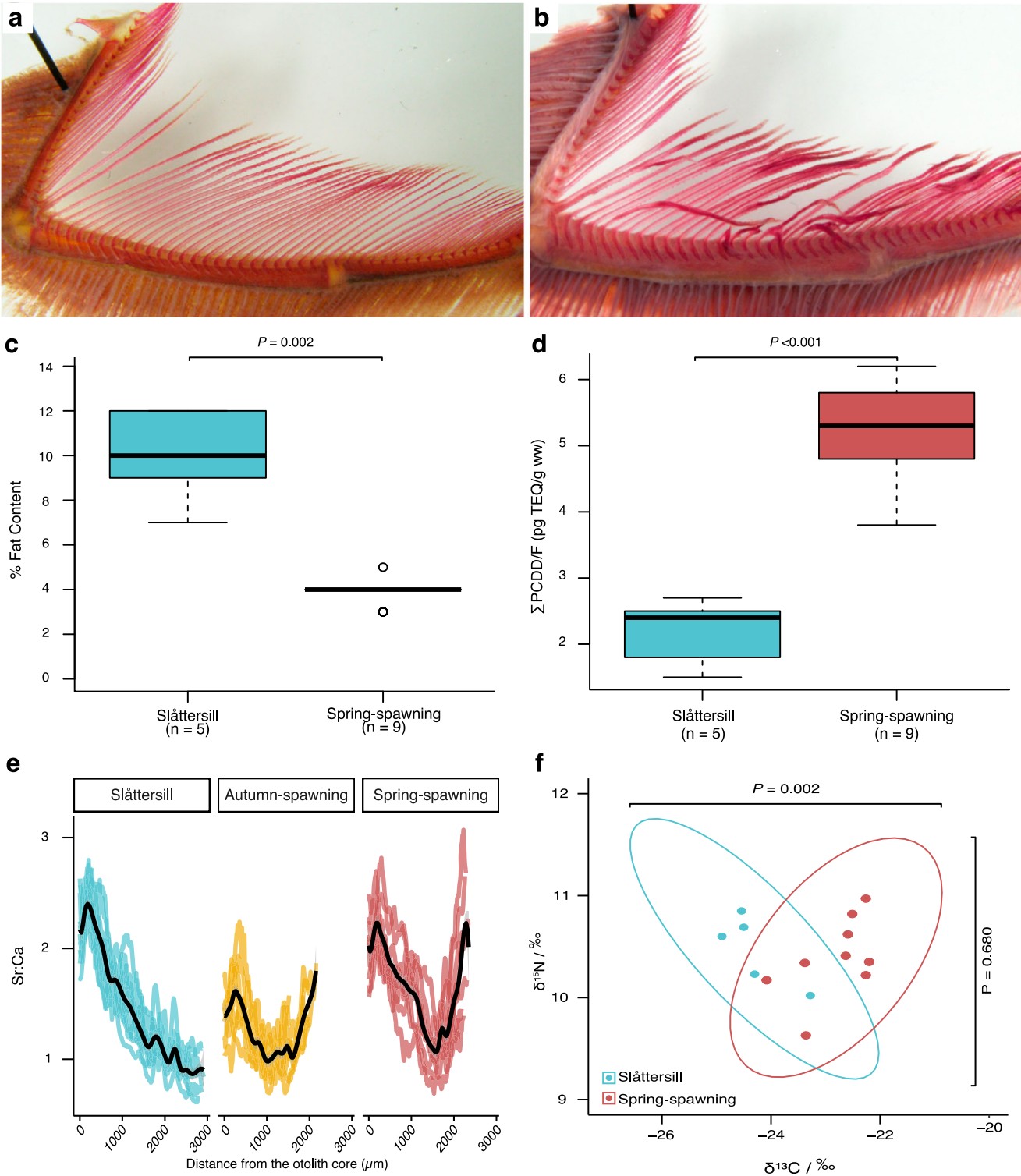

**Fig. 2 | Phenotypic comparison of Slåttersill and small, spring-spawning Baltic herring. a** Undamaged gill rakers from planktivorous spring-spawning Baltic herring. **b** Gill rakers with lesions frequently observed in Slåttersill most likely caused by a fish diet. **c** Fat content (%) in skeletal muscle from Slåttersill ($n = 5$ pools of samples representing 96 fish in total) and small, spring-spawning Baltic herring ($n = 9$ pools of samples representing 170 fish in total). The box-plot shows the median value as the center point, while the box spans the interquartile range (IQR) between Q1 and Q3, with whiskers indicating the maximum spread of data points within 1.5 times the IQR from Q1 and Q3. **d** Sum of polychlorinated dibenzo-p-dioxins (PCDD) and polychlorinated dibenzofurans (PCDF) measured in picograms of toxic equivalent per gram of wet weight per sample (∑PCDD/F (pg TEQ/g ww)),

respectively, sampled from Slåttersill ($n = 5$ pools of samples representing 96 fish in total) and spring-spawning Baltic herring ($n = 9$ pools of samples representing 170 fish in total). The box-plot shows the median value as the center point, while the box spans the IQR between Q1 and Q3, with whiskers indicating the maximum spread of data points within 1.5 times the IQR from Q1 and Q3. **e** A comparison of lifelong otolith strontium:calcium profiles of Slåttersill (blue), autumn-spawning (yellow), and spring-spawning Baltic herring (red) from Gävlebukten with smoothed splines in black. **f** Stable isotope analysis of the same individuals as in **d** for carbon-13 to carbon-12 ratio relative to a standard (δ¹³C/‰), and for nitrogen-15 to nitrogen-14 ratio relative to a standard (δ¹⁵N/‰), expressed in parts per thousand. Source data are provided as a Source Data file.

truly piscivorous Wolf-herrings (*Chirocentrus* spp.) serves as an example, with the latter possessing a lower number, of short and sparsely distributed gill rakers[23]. Thus, a plausible interpretation of the lesions in Slåttersill is a recent switch from zooplankton to a fish diet, not yet accomplished with genetic adaptation for piscivory.

A change in diet may affect the accumulation of pollutants in fish, and we, therefore, decided to examine this as a test for a changed diet in the Slåttersill population. Pollution from chlorinated organic compounds, e.g., polychlorinated dibenzo-p-dioxins and polychlorinated dibenzofurans (PCDD/Fs; dioxins), has been an environmental problem of concern in the Baltic Sea for decades[24-26]. Being fat-soluble, the dioxin content in fish generally correlates well with the lipid content when controlling for other factors, e.g., age. Surprisingly, Slåttersill, while having significantly higher lipid content than the spring-spawning herring ($\bar{x}\text{-}\bar{y} = 6.1\%$, C.I. = 3.5–8.7%, two-sided $t$ test, $t(4) = 6.3$, $P = 0.002$; Fig. 2c), simultaneously had significantly lower PCDD/F levels in muscle and skin tissue ($\bar{x}\text{-}\bar{y} = -3.0$ pg TEQ/g ww, C.I. = −2.3−−3.8 pg TEQ/g ww, two-sided $t$ test, $t(12) = -8.6$, $P = 2.3 \times 10^{-6}$; Fig. 2d, Supplementary Table 4). As PCDD/Fs have a strong affinity to particles and accumulate in sediments, impacting benthic organisms in particular, a possible explanation is differences in the diet between ecotypes. Small planktivorous herring from the studied area, at least for parts of the year, feed on deep water macro-crustaceans, i.e., *Mysis*, containing relatively high levels of PCDD/Fs[27]. The much larger Slåttersill, as discussed above, most likely feeds on fish at larger sizes, likely being less exposed to the sediment-bottom water accumulation of PCDD/Fs. Moreover, the average age of Slåttersill was significantly lower compared to the small, spring-spawning herring (see above, Fig. 1c, Supplementary Table 2), implying less time for PCDD/F to bioaccumulate, which also may have influenced the difference in PCDD/F levels[28]. However, levels of polychlorinated biphenyls (PCB), another bioaccumulating organohalogen of importance for the Baltic Sea, did not deviate between the two groups ($\bar{x}\text{-}\bar{y} = 7.2$ ng/g ww, C.I. = −1.9−16.4 ng/g ww, two-sided $t$ test, $t(7) = 1.85$, $P = 0.10$; Supplementary Table 4), making age differences likely less critical for the observed PCDD/F anomaly.

Next, we quantified the ratio of strontium (Sr) and calcium (Ca) in otoliths as a proxy for salinity[29], to assess migration patterns. The results indicated lifelong ecotype-specific migration strategies (Fig. 2e). Pairwise comparison of otolith log-transformed Sr:Ca slopes showed a general decline with age in adult Slåttersill, significantly different from the increasing trends in spring- and autumn-spawning herring from Gävlebukten (Fig. 2e, Supplementary Fig. 4, and Supplementary Table 5). Adult Slåttersill otolith Sr:Ca profiles displayed both overall lower levels and less seasonal variations than spring- and autumn-spawning Baltic herring, suggesting that Slåttersill utilize habitats with lower salinity and that Slåttersill does not make long-distance migrations (Fig. 2e). Thus, we conclude that the low levels of dioxin in Slåttersill are not caused by long-distance migration to feeding grounds with reduced contamination in the southern Baltic Sea, but is most likely explained by an altered diet and faster growth rate compared with small Baltic herring.

The altered growth rate, the damage on the gill rakers, and the significant reduction in dioxin contamination all implied an altered feeding behavior and, most likely, a switch to a more nutrient-rich fish diet. To test this hypothesis, we performed a stable isotope analysis of the $\delta^{15}N$, the ratio of $^{15}N/^{14}N$, which shows a positive correlation with the trophic level[30,31]. The comparison of Slåttersill and spring-spawning herring revealed no significant difference in $\delta^{15}N$ ($\bar{x}\text{-}\bar{y} = 0.08‰$, C.I. = −0.37−0.54‰, two-sided $t$ test, $t(9) = 0.43$, $P = 0.68$; Fig. 2f, Supplementary Table 4). However, $\delta^{15}N$ in the consumer tissue is also influenced by the protein content of its diet and by the growth rate of the consumer, both of which can lower the so-called trophic discrimination factor for $^{15}N$ and result in lower $\delta^{15}N$ values[32,33]. Thus, the fact that these two populations show a highly significant difference in

growth rate (Fig. 1c) makes us suggest that Slåttersill, despite its similar $\delta^{15}N$ ratio, is at a higher trophic level than the small spring-spawning herring, consistent with a switch to a high protein fish diet.

## Whole-genome sequencing reveals the presence of multiple subpopulations of piscivorous Baltic herring

The genetic constitution of the Slåttersill population was further characterized by WGS because the SNP chip used here was designed prior to the discovery of this genetically unique population. We included six additional population samples of unusually large non-spawning Baltic herring from different parts of the Baltic Sea (Fig. 1a, Supplementary Table 6) to test if these represent the same population as Slåttersill. The stomach content of these large Baltic herring had previously been characterized[34], and was found to be dominated by fish. Interestingly, the dominating prey (>60%) was three-spined stickleback (*Gasterosteus aculeatus*), which is highly abundant in the Baltic Sea and carries sharp spines that could explain the lesions noted on the gill rakers of Slåttersill (Fig. 2b).

Individual libraries for WGS were prepared from 103 Baltic herring (Supplementary Table 6) using Tn5 tagmentation[35] and sequenced to an average 11.8× (±2.8) sequence coverage. A PCA analysis comparing the allele frequencies of seven population samples of these large piscivorous Baltic herring, including Slåttersill, with previously described ecotypes of Atlantic and Baltic herring[14] revealed that the large Baltic herring clustered with spring-spawning Baltic herring or with some spring-spawning populations from the transition zone that connect the Atlantic Ocean with the Baltic Sea (Fig. 3a). Next, we made a genome-wide comparison of each large Baltic herring with the average allele frequencies observed in previously sequenced small, spring-spawning Baltic herring[14]. This analysis underscored differences between small and large herring and also demonstrated that the latter is genetically heterogeneous and comprises multiple subpopulations (Fig. 3; Supplementary Fig. 5). First, we identified a striking signal of genetic differentiation in a 3.9 Mb region on chromosome 20 (12.8–16.7 Mb) in five of the seven large herring populations (Fig. 3b–d; Supplementary Fig. 5c–g). This signal was present in Slåttersill but not detected in the SNP-chip analysis above since no SNPs from this region were present on the SNP-chip. The signal at chromosome 15 noted in the SNP-chip analysis of Slåttersill (Fig. 1e) was only replicated in the large herring from Gävlebukten, close to the area where Slåttersill was sampled (Supplementary Fig. 5c). The sharp signal on chromosome 19 (Fig. 1e), on the other hand was noted in all large herring except in the sample from Blekinge (Supplementary Fig. 5b). The most striking example of genetic heterogeneity among the large Baltic herring was the genetic differentiation for the 7.8 Mb supergene (inversion) present in three population samples, Stockholm, Östergötland, and Kalmar (Fig. 3e, f; Supplementary Fig. 5e–g).

In conclusion, WGS of seven population samples of large piscivorous herring reveals genetic differentiation compared with the much more abundant small planktivorous Baltic herring, but the piscivorous herring are composed of at least two distinct subpopulations. One is present in the Bothnian Sea, represented by Slåttersill and the sample from Gävlebukten (denoted northern piscivorous), and the other one was found in the central Baltic Sea and represented by the samples from Stockholm, Östergötland, and Kalmar (denoted southern piscivorous). Further work is required to determine if the population samples from Blekinge and Finland reflect additional genetically distinct populations of large piscivorous Baltic herring.

## Similar levels of nucleotide diversities among Baltic herring ecotypes but variable patterns of reproductive isolation

To further assess patterns of allele sharing, gene flow, and historical demographic trends, we grouped herring according to the four major ecotypes of the study region (northern vs. southern large piscivorous herring and autumn- vs. spring-spawning small planktivorous herring).

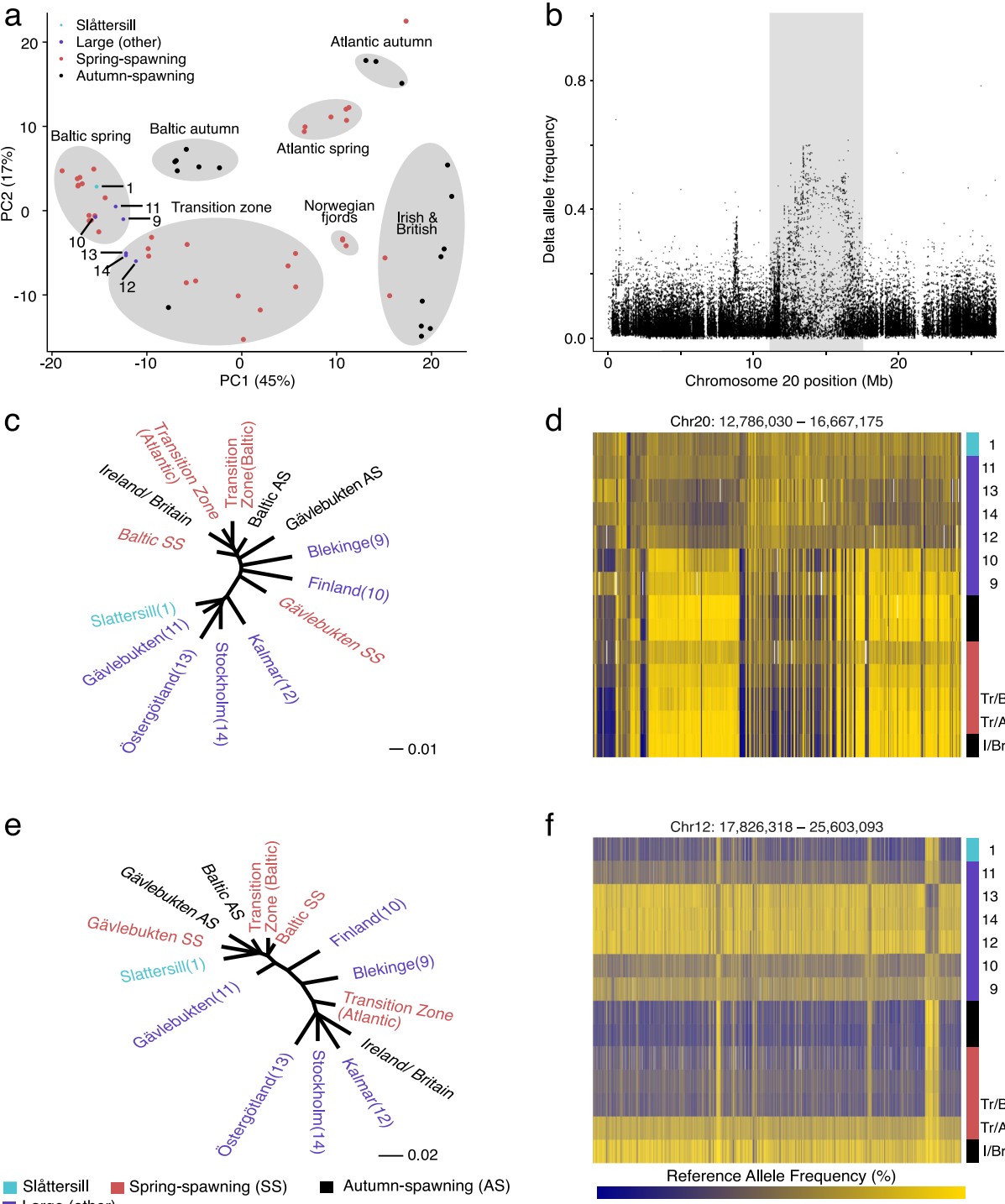

**Fig. 3 | Whole-genome sequencing reveals genetic heterogeneity among large Baltic herring.** Comparison of Slåttersill and six other population samples of exceptionally large Baltic herring relative to previously sequenced Atlantic and Baltic herring samples[14]. **a** PCA plot of WGS data filtered to 407 (out of 794) markers previously used to define primary population groups in Atlantic herring[14]. The seven primary population groups are represented as labeled gray ellipses, while the individual points represent pooled per-location averages. The color of individual points represents the group each represents, i.e., Slåttersill (blue), large Baltic herring (other than Slåttersill, purple), spring-spawning herring (red), and autumn-spawning herring (black). **c** Neighbor-joining tree based on allele frequency differences for the chr20 region of interest (12.8–16.7 Mb) for all large Baltic herring, small spring- and autumn-spawning Baltic herring, and populations from the Transition Zone (Baltic

side, Tr/B; Atlantic side, Tr/A) and Irish and British (I/Br) waters. **d** Zoom-in heatmap of the chr20 region of interest (12.8–16.7 Mb) highlighted in the gray box in Fig. 3b using the same population samples as in Fig. 3c (delta allele frequency cutoff >0.25). **e** Neighbor-joining tree based on allele frequency differences for the chr12 inversion region (17.8–25.6 Mb) using the same population samples as in Fig. 3c. **f** Zoom-in heatmap of the chr12 inversion region (17.8–25.6 Mb) using the same population samples as in Fig. 3c (delta allele frequency cutoff >0.35). Population samples of large herring are labeled as follows: Slåttersill (1); Blekinge (9); Finland (10); Gävlebukten (11); Kalmar (12); Östergötland (13); Stockholm (14) (see Supplementary Table 6). Samples sourced from outside of the Baltic Sea are labeled using letters, where: Transition Zone (Baltic side, Tr/B), Transition Zone (Atlantic side, Tr/A), Ireland/Britain (I/Br). Source data are provided as a Source Data file.

We then used the WGS data to estimate nucleotide diversity (Watterson's $\theta$ and $\pi$ per base), Tajima's $D$, and long-term effective population sizes ($N_e$) for each group (Supplementary Table 7). The data show very similar levels of diversity ($\theta_W$ = in the range 0.0045–0.0050). This is a somewhat surprising result given that it is obvious that planktivorous herring is much more abundant than piscivorous herring in the Baltic Sea[17]. It implies that the establishment of piscivorous populations of Baltic herring was not associated with a strong founder effect, or that such a founder effect has subsequently been erased by gene flow. We performed pairwise estimates of genetic differentiation ($F_{ST}$) among the ecotypes and estimated $F_{ST}$ values to be in the range of 1.6–2.6%, with the large herring clustering with spring-spawners (Supplementary Fig. 6a). We found that the northern piscivorous herring (Slåttersill) shared much variation with spring-spawning planktivorous herring, as it showed lower genetic divergence to spring-spawners than any other ecotype across 47% of the genome and at 21 out of 26 chromosomes (Supplementary Fig. 6b, c). In contrast, the southern piscivorous herring appeared to share considerably less variation with spring-spawners, being most similar to the northern large herring (Supplementary Fig. 6b–c). These observations suggest that the northern and southern piscivorous populations have a common ancestry but the former may have exchanged more genetic material with planktivorous herring than the southern herring, which instead have been more reproductively isolated.

### A unique recombinant version of the Chr12 supergene in herring

One of the most important loci for local adaptation in herring is the 7.8 Mb supergene inversion on chromosome 12, for which haplotypes segregate according to latitude and temperature at spawning among populations[14,36]. Interestingly, this locus has diverged also among northern and southern piscivorous herring in the Baltic Sea. We found that the three piscivorous population samples (Stockholm, Östergötland, and Kalmar) representing the southern ecotype were almost fixed for a haplotype closely related to the Southern haplotype of the chromosome 12 supergene (Fig. 3e, f; Supplementary Fig. 5e–g). This haplotype is close to fixation in populations spawning in the waters surrounding Ireland and Great Britain, which are the warmest waters where herring spawning occurs. A comparison of southern piscivorous herring against spring-spawning planktivorous Baltic herring[14], not carrying this haplotype, reveals the classical pattern for an inversion with strong genetic differentiation across the inverted region and with supersharp borders between SNPs inside and outside the inversion (Fig. 4a), whereas a comparison against herring from Ireland/Great Britain did not reproduce this inversion pattern (Fig. 4b), suggesting that the latter pair share very similar haplotypes. However, a closer inspection reveals three peaks of striking genetic differentiation within the inversion (marked c, d, and e in Fig. 4b). Region **c** apparently represents a recombination between the Northern and Southern haplotypes because the Southern haplotype present in large herring shares 12 diagnostic SNPs with Northern haplotypes (Fig. 4c); the differentiation in flanking SNPs most likely represents a selective sweep subsequent to the recombination event. Similarly, region **e** also represents a recombination between a Northern and Southern haplotype as indicated by many diagnostic SNPs (Fig. 4e), while region **d** may represent a selective sweep that has occurred on the Southern-like haplotype present in the southern piscivorous population (Fig. 4d). The presence of this unique version of the Southern haplotype, at a frequency of -0.8, which is rare or absent in other populations of Atlantic and Baltic herring (Fig. 4c–e), provides evidence that the southern piscivorous population is a genetically unique population of herring not previously described.

## Discussion

Sympatric differentiation and niche expansion are well documented in freshwater fish, in particular salmonids[5] and in cichlids[37]. There are two reasons why the discovery of sympatric genetic differentiation and niche expansion in Baltic herring was unexpected. Firstly, the Atlantic herring is a specialized planktivorous fish of huge importance for the ecosystem in the North Atlantic Ocean. It is, therefore, unexpected that this species shows niche expansion and evolution of a piscivorous ecotype after colonizing the brackish Baltic Sea. Secondly, the Atlantic and Baltic herring are highly abundant broadcast spawners that release eggs and sperm into seawater, which means there is no mate choice that can facilitate reproductive isolation. In contrast, non-genetic imprinting, for instance, song, is important for mate choice in Darwin's finches and other birds, which may contribute to the establishment of reproductive isolation and subsequent genetic differentiation[38,39].

The Atlantic herring, as well as the small Baltic herring, are typical plankton-feeding fish, both displaying gill raker morphology consistent with this feeding behavior (Fig. 2a). Atlantic herring in proper marine environments may occasionally feed on fish eggs and larvae[40,41], but extensive feeding on larger fish has not, to the best of our knowledge, been described outside the Baltic Sea. The common small Baltic herring feed on zooplankton (90%), mysids, and benthos[42], while it has been reported that unusually large Baltic herring may feed on fish[16,17]. However, it was unknown if these herring simply represented the right-most end of the size distribution, and that Baltic herring of a specific size may switch feeding behavior. This study demonstrates that these herring constitute genetically differentiated ecotypes with distinct growth and migratory patterns (Figs. 1 and 2e and Supplementary Fig. 4).

The finding that herring shows niche expansion in the Baltic Sea, but most likely not in the Atlantic Ocean, likely reflects reduced competition from other piscivorous fish in the Baltic Sea. The Baltic Sea is a young water body formed within the last 8000 years, after the end of the last glaciation[11]. Few marine fish have colonized the Baltic Sea, and large predators such as different species of mackerel, tuna, and pelagic gadoids are not present. Several other species of fish show adaptation to the Baltic Sea, such as Atlantic cod[43], which has differentiated strongly from its conspecifics in fully marine environments, and the endemic Baltic flounder[44]. However, the ecotypes of herring described here appear to be unique by showing both sympatric genetic differentiation and niche expansion.

We did not notice significant genetic differentiation at neutral loci in the piscivorous Baltic herring despite the striking genetic differentiation at dozens of adaptive loci (Figs. 1e, 3 and 4), suggesting low genetic drift and incomplete reproductive isolation, although to a different degree between the northern and southern piscivorous populations. Population genetics theory has established that very little gene flow (one migrant per generation) is sufficient to avoid differentiation and loss of heterozygosity in subdivided populations[45]. The level of gene flow that has eliminated differentiation at neutral loci has not prevented the sympatric evolution of piscivorous Baltic herring, implying that this ecotype has evolved through strong natural selection. What makes the evolution of piscivorous Baltic herring unique is that this niche expansion has not been reported in other parts of the species distribution, and the conundrum of why the piscivorous ecotypes have not been genetically absorbed by the abundant planktivorous Baltic herring with an estimated census population size of -70 billion individuals[6].

How is sympatric genetic differentiation possible in an extremely abundant fish that makes long-distance migrations, and is a broadcast spawner? Our recent description[46] of a long-standing hybrid population formed by hybridizations between the two sister species, Atlantic

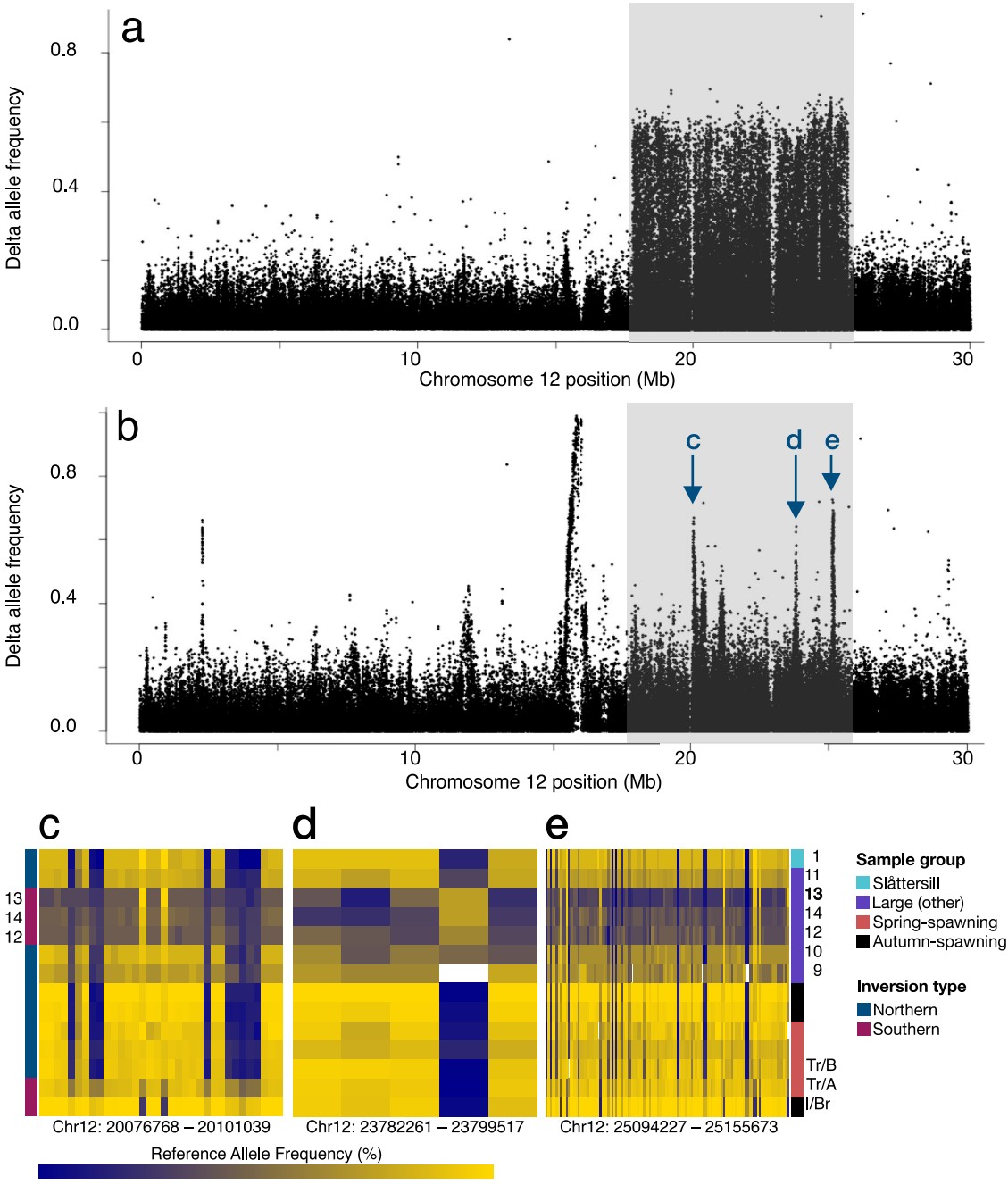

**Fig. 4 | A recombinant version of the chromosome 12 supergene. a** Delta allele frequency contrast between pooled allele frequencies for three large herring samples (Östergötland, Stockholm, and Kalmar) and small, spring-spawning Baltic herring. The chromosome 12 inversion region (12:17.8–25.6 Mb) is highlighted by the gray box. **b** Delta allele frequency contrast between pooled allele frequencies for three large herring samples (Östergötland, Stockholm, and Kalmar) and Atlantic herring from the Transition Zone (Baltic side, Tr/B; Atlantic side, Tr/A) and Irish and British waters. Significant peaks of differentiation within the chr12 inversion region (gray box) are annotated relative to the subsequent zoom-in heatmaps used to visualize each peak. **c**–**e** Zoom-in heatmaps of chr12: 20,076,768–20,101,039 bp region (**c**), chr12: 23,782,261–23,799,517 bp region (**d**), and chr12:

25,094,227–25,155,673 bp region (**e**). The color code to the left of (**c**) indicates if the predominant inversion haplotype is classified as southern or northern[14]. The population samples are colored as follows: Slåttersill (blue), other large Baltic herring (purple), spring-spawning herring (red), and autumn-spawning herring (black). The labeling of large Baltic herring samples is as in Fig. 3: Slåttersill (1); Blekinge (9); Finland (10); Gävlebukten (11); Kalmar (12); Östergötland (13); Stockholm (14) (see Supplementary Table 6). Samples sourced from outside of the Baltic Sea are labeled as follows: Tr/B (Transition Zone, Baltic side), Tr/A (Transition Zone, Atlantic side), and I/Br (Ireland/Britain). Source data are provided as a Source Data file.

herring and Pacific herring (*Clupea pallasii*), implies that there is not any strong prezygotic isolation in herring, as expected for a broadcast spawner lacking mate choice. The fact that this Pacific/Atlantic hybrid population has existed for thousands of years without being absorbed by the huge population of Atlantic herring present in the same

geographic area is of interest in relation to the ability of the piscivorous populations of Baltic herring to maintain their unique genetic profile. This occurs even though they are spawning in the same geographic region during the same period as spring-spawning planktivorous herring. The implication is either that the herring has a more

sophisticated homing behavior than previously thought[47], or, alternatively, there exists a spawning behavior that restricts gene flow from other populations. A strong homing behavior is adaptive in herring because it allows genetic adaptation to environmental conditions at spawning. The herring deposits the fertilized eggs on vegetation or on the sea floor, and these are thereby exposed to the environmental conditions on the spawning grounds to a much larger extent than a pelagic spawner where the fertilized eggs drift with sea or ocean currents. Further, there are very strong environmental gradients (e.g., temperature, salinity, water transparency, and primary production) in the Baltic Sea coastal zone, especially during spring[48], which would underscore the importance of a pronounced homing behavior for populations to optimize chances of successful reproduction.

The main prey of the piscivorous Baltic herring appears to be the abundant three-spined stickleback[34], a species with sharp spines known to cause damage to its predators[49]. Similar lesions have been documented in fish-eating kokanee salmon (land-locked *Oncorhynchus nerka*) that also had stickleback as its main prey. While morphological, genetic, and stable isotope techniques did not support the separation of piscivorous and planktivorous kokanee into discrete ecotypes, it was suggested that an increase in gill raker spacing due to a larger body size of piscivorous individuals, in combination with damage on gill rakers, would reduce foraging efficiency on zooplankton and thus reinforce the shift to piscivory on the individual level[50]. A similar phenotypic plasticity and feedback mechanism may have been the initial step for the evolution of the piscivorous herring ecotype, followed by a certain degree of reproductive isolation due to strong homing behavior and natural selection.

The discovery that a genetically distinct subpopulation of large, fish-eating Baltic herring is close to fixation for a unique recombinant version of the Southern haplotype for the chromosome 12 inversion demonstrates that this supergene undergoes dynamic evolution. This is consistent with a detailed characterization of the four major supergenes in herring (on chromosome 6, 12, 17, and 23) based on PacBio long-read sequencing that reveals that these originated more than a million years ago and that there is a considerable gene flux between inversion haplotypes at all four loci[51]. A similar example of gene flux between inversion haplotypes has recently been reported in the long-snouted seahorse[52]. The four supergenes play a prominent role in ecological adaptation in Atlantic and Baltic herring, and the supergenes on chromosome 17 (Fig. 1e) and chromosome 12 (Fig. 4) have both contributed significantly to the genetic differentiation of the piscivorous Baltic herring. The Southern haplotype is uncommon at higher latitudes in both the Baltic Sea and the Atlantic Ocean[14,36], hinting at parallel sorting of adaptive alleles at this locus between ocean basins.

The current rapid loss of biodiversity and genetic diversity due to human expansion and exploitation of natural resources are of major concern[53]. This study demonstrates that also in extremely abundant species, precious and potentially vulnerable genetic diversity may be present, although not detectable using the standard approach for monitoring genetic diversity using a sparse set of selectively neutral markers. There is an obvious risk that the currently high fishing pressure on Baltic herring populations may lead to losses of genetic diversity and unique subpopulations, warranting cautious management. This could be particularly important for the Baltic Sea, where the ecosystem depends on a few species and trophic interactions are easily disturbed. For example, perch and pike normally prey on coastal three-spined stickleback but have recently declined, leading to a massive increase in sticklebacks that instead feed on the larvae of their predators (so-called predator–prey reversal), limiting their recruitment and reinforcing trophic regime shifts in the Baltic Sea[54]. If managed favorably, the large piscivorous herring characterized here could help control the stickleback population and be an unexpected and valuable ally in restoring Baltic food webs. A comprehensive understanding of the population structure is particularly important in species that are heavily exploited by industrial fishing[18], both for avoiding unsustainable overfishing and for protecting ecosystem function. This includes the Atlantic and Baltic herring, which sustain one of the top ten most important fisheries in the world[55].

## Methods
Fish samples included in this study were collected from commercial herring fishery for which no ethical permission is required and from scientific surveys in compliance with the EU Directive 2010/63/EU and Swedish national legislation, under permits C 139/13 and 5.8.18-10169/2019, issued to the Swedish University of Agricultural Sciences by the Regional Ethical Committee on Animal Experiments.

### SNP-chip analysis
Local fishers were chartered for the capture of fish from eight locations across Gävlebukten, Sweden (Supplementary Table 1). Charters were conducted in 2022 and 2023, between April and October to capture both Slåttersill ($n = 107$), as well as spring- ($n = 201$) and autumn-spawning Baltic herring ($n = 79$). Length and weight were recorded before collecting muscle tissue using a Biomark Tissue Sampling Unit (Biomark LLC), then stored at −80 °C.

DNA extraction and the following SNP-chip analysis were performed by IdentiGEN (Dublin, Ireland) using the MultiFishSNPChip_1.0 array (FSHSTK1D)[18]. The herring component of the SNP array was designed to cover all independent regions of divergence that had been identified at the time, including a large proportion of missense mutations showing strong genetic differentiation between populations. The array also includes a set of neutral markers, defined as having minor allele frequency above 0.3 but little inter-population variation spread approximately evenly along the chromosomes. The full list of SNP designs is provided as Supplementary Data 3.

The genetic spawning season for all fish was predicted using AssignPop[56]. All SNP-chip genotyped fish were compared against an Atlantic herring reference database comprising $n = 8$ individuals simulated from per-location ($n = 53$) pooled allele frequency data[14]. Each of the 53 reference pools was used previously to define the seven population clusters inherent to the species' North Atlantic range (i.e., Baltic Sea (Autumn), Baltic Sea (Spring), Atlantic Ocean (Autumn), Atlantic Ocean (Spring), Ireland and Britain, Norwegian Fjords, and Transition Zone)[14], with the current Atlantic herring reference database reflecting these assignments. All SNP-chip genotyped fish were therefore assigned to one of the seven population clusters in AssignPop, with genetic spawning season characterized only when individuals were assigned to either the Baltic Sea (Autumn), Baltic Sea (Spring), Atlantic Ocean (Autumn), or Atlantic Ocean (Spring) groups. Population assignment was deemed successful when AssignPop assignment probabilities were >66% for a single population cluster (i.e., two-thirds more likely than the next alternative outcome[57]).

PCA was carried out using function (--pca) in PLINK2[58]. The MultiFishSNPChip_1.0 array contains a high redundancy of SNPs within inversions, showing strong linkage disequilibrium. Therefore, PCAs were both calculated, including all available SNPs, or a dataset in which the inversions were treated as single loci. To achieve the latter, all individuals were genotyped for the inversion, by comparing SNP data against reference haplotype assignments sourced from Han et al.[14]. Reference haplotypes constituted the consensus allele for either the Northern (*N*) or Southern (*S*) haplotype derived from pure Northern (Norway Spring, Iceland Spring, and Greenland Summer) or pure Southern haplotype (Isle of Man Autumn, Celtic Sea Autumn, and Downs Winter), respectively. Each SNP within inversion was scored against the reference set, and genotyped as *NN*, *NS*, or *SS*. The genotype for each inversion was then deduced (*NN, NS,* or *SS*) consistent with the predominant haplotype across each inversion. As the spawning season is expected to be the predominant signal differentiating populations of Baltic herring, individuals were visualized in R,

and assignments to spawn season were predicted using K-means clustering ($K = 2$) of PLINK-derived eigenvalues.

Allele frequencies were calculated using PLINK2[58] for all population samples independently, as well as for groups of spring-spawning Baltic herring, autumn-spawning Baltic herring, and Slåttersill fish groups. PCA was calculated in R (*prcomp* function) and visualized using ggplot2[59]. Genotype BED files and association analyses were generated with PLINK (v1.9), and Bonferroni corrected *P* values calculated using the --assoc and --adjust functions, then visualized using R. Gene annotation for all SNP was sourced from BioMart[60] using the 'char-engus_gene_ensembl' dataset as a reference.

### Whole-genome sequencing (WGS)

The population samples of exceptionally large Baltic herring included, in addition to Slåttersill, five population samples from Sweden and one from Finland (Supplementary Table 6). A control sample (Gävleborg Norrsundet) was also collected, representing small, spring-spawning Baltic herring from the same geographic region as where Slåttersill occurs. All fish were sampled between 2020 and 2022, with fish length recorded before muscle tissue was collected by dissection and stored at −80 °C.

A total of 86 individuals were used to prepare WGS libraries with Tn5-based tagmentation. The method for library preparation was an implemented version of a previous protocol used in our laboratory[61] adapted from Picelli et al.[35]. DNA was extracted with the DNeasy 96 Blood & Tissue Kit (Qiagen, Hilden, Germany) and quantified with the NanoDrop One C (Thermo Fisher Scientific, Madison, Wisconsin, USA). Genomic DNA from Slåttersill was extracted at IdentiGEN Ltd. (Ireland) as part of the SNP-chip analysis and consisted of a crude extract obtained using Chelex. The DNA integrity for all samples was checked on a gel. The samples were then diluted to 10 ng/μL and 2 or 4 μL were used for library preparation. Tn5 tranposase (Tn5 Tnp) was purchased from the Protein Science Facility at the Karolinska Institute (Stockholm, Sweden). The enzyme was loaded with pre-annealed mosaic end primer pairs, ME-A (Fw: 5′-TCGTCGGCAGCGTCAGATGTGTATAAGA-GACAG-3′ and R: 5′-CTGTCTCTTATACACATCT-3′) and ME-B (Fw: 5′-GTCTCGTGGGCTCGGAGATGTGTATAAGAGACAG-3′ and R: 5′-CTGTCTCTTATACACATCT-3′), for 2 h at room temperature. The complex was diluted to a final concentration of 8 or 6.4 ng/μL in the tagmentation reactions. Tagmentation was allowed to proceed for 5 to 10 min at +55 °C, then Tn5 transposon was stripped off the DNA template with 0.04% SDS. Primer sequences for library indexing were from Illumina Nextera Library Prep Kits and were combined to give a unique combination of i5 and i7 indexes per sample. The primers were appended to the DNA molecules by a 9-cycle PCR with the KAPA HiFi PCR Kit (Kapa Biosystems Pty, Cape Town, South Africa) with cycling parameters of 72 °C for 3 min for gap-filling and 98 °C for 30 s for denaturation, followed by 9 cycles of 98 °C for 30 s, 63 °C for 30 s, 72 °C for 3 min. A double-sided size selection of 0.5×−0.7× was performed with AMPure XP paramagnetic beads (Beckman Coulter, Inc., Brea, California, USA)[62]. An aliquot of each individual library was mixed with 70 μL of Qubit HS reagents (Life Technologies Corporation, Eugene, Oregon, USA) prepared according to instructions, and fluorescence was read with a TECAN infinite M200 microplate reader (Tecan Austria GmbH, Grödig, Austria). The Qubit controls were included in the measurements to construct a linear regression line and calculate the individual library concentrations. Three ng were pooled per library, and the pool was concentrated to 35 μL with another round of 0.5×−0.7× size selection. The insert size was checked with a TapeStation 4150 (Agilent Technologies, Waldbronn, Germany), and the concentration was measured by qPCR with the KAPA Library Quant Kit (Kapa Biosystems Pty, Cape Town, South Africa). Sequencing consisted of paired-end sequencing with 150 bp read length and was performed by the SNP&SEQ Technology Platform in Uppsala on a NovaSeq 6000 with a S4 flowcell and v1.5 sequencing chemistry.

Raw paired-end reads were mapped to the *Clupea harengus* reference assembly[36] (Ch_v2.0.2.fasta) using samtools[63]. Variants were called from BAM files using a standard gatk pipeline[64]. Firstly, BAMs were indexed using samtools, before individual gvcf were generated using the *gatk HaplotypeCaller* function. All sample-specific gvcf were combined using *CombineGVCFs* and genotypes called using *GenotypeGVCFs*. SNP and INDELs were subset into independent variant files using *SelectVariants*, with the SNP-specific subset further filtered for biallelic SNP and multiple expression using *VariantFiltration* (QD < 2; QUAL < 30; SOR > 3; FS > 60; MQ < 40; MQRankSum < −12.5; Read-PosRankSum < −8) and bcftools/1.17. Pooled allele frequencies were calculated independently using PLINK2[58] for all populations.

Allele frequencies per sample group were calculated from the called genotypes using the "freq2" output from vcftools, v0.1.16[65], and delta allele frequencies were then calculated using R v3.6[66]. For plotting purposes, only sites with at least 140 out of a possible 152 (92%), observed chromosomes were retained. SNPs were imputed and phased with BEAGLE v4[67] ahead of the estimation of levels of genetic diversity and divergence among ecotypes. Samples were then grouped into ecotypes according to location, morphology, and spawning genotypes (Supplementary Table 7, Supplementary Data 4). Levels and patterns of diversity ($\theta_W$, π, and Tajima's D) were estimated across the 26 chromosomes using a custom Perl script that incorporated BioPerl code for Tajima's D[68,69]. The standard equation $N_e = \theta_W/4\mu$ was used to infer effective population size using per base $\theta_W$ estimates and the herring mutation rate of $2.0 \times 10^{-9}$ per base and generation[6]. Reynold's $F_{ST}$[70] was used to estimate pairwise divergences among ecotypes across the whole-genome and in 10 kb windows, and the UPGMA clustering algorithm was used to build a population tree from genome-wide estimates, as implemented in the neighbor program in PHYLIP v3.697[71].

### Otolith analysis

The individual ages of samples of Slåttersill ($n = 32$), spring-spawning Baltic herring ($n = 31$), and autumn-spawning Baltic herring ($n = 26$) were estimated by counting annuli on transverse otolith sections stained with toluidine blue.

For otolith chemistry analyses, otoliths of Slåttersill ($n = 10$), spring-spawning Baltic herring ($n = 10$), and autumn-spawning Baltic herring ($n = 8$) were first embedded in epoxy, then hand-grinded and polished to the sagittal midplane to expose the core. Otolith trace elemental concentrations of strontium ($^{88}$Sr) and calcium ($^{43}$Ca) were analyzed with laser ablation inductively coupled plasma mass spectrometry (LA-ICP-MS) at the College of Environmental Science and Forestry at the State University of New York (SUNY-ESF) in Syracuse, New York, USA. Otolith material was ablated along a line transect from the core (birth) to the anterior rostrum edge (death) following the maximum growth axis (Supplementary Fig. 4a). The otoliths were analyzed with a round laser spot size of 85 μm at a scanning speed of 5−6 μm/sec with a repetition rate of 10 Hz, using helium and argon as carrier gas. The reference material MAPS-5 was used as the primary standard, and NIST SRM 612 was used for tuning and checking the daily performance of the instrument. Iolite, Excel, and R were used for data reduction and statistical analyses with $\alpha < 0.05$ as the threshold for statistical significance[72–74].

Otolith chemical profiles as regards the Sr:Ca ratio (a proxy for salinity[29]) were assessed for comparing lifelong migration patterns of the different ecotypes; Slåttersill, spring-spawning Baltic herring, and autumn-spawning Baltic herring. Repeated measurements of Sr:Ca along the line transect of each otolith were used for performing mixed-effects regression analyses (lmer) with the R package lme4[75]. The otolith Sr:Ca ratio was log-transformed to better fit model assumptions and attain symmetrically distributed residuals. Ecotype, distance from the otolith core (μm), and the interaction term were used as fixed factors, and sample ID as the random factor (model formula: log

Sr:Ca ~ ecotype + distance + (ecotype * distance) + (1|sample ID)). No multicollinearity between predictor variables was detected (variance inflation factor (VIF) < 5). Statistical pairwise comparisons of migration patterns of adult Slåttersill, spring-spawning Baltic herring, and autumn-spawning Baltic herring were performed by comparing the model slopes of log (Sr:Ca) at ≥1500 μm from the otolith core corresponding to adult fish using the emtrends function in the emmeans R package[76]. The R package ggplot2 was used for graphical comparisons of spawning types[59].

### Morphological analysis

A total of 47 specimens were analyzed including 10 specimens each of Slåttersill and Baltic spring-spawning herring (Supplementary Data 2). The material was received and frozen at the Swedish Museum of Natural History (NRM), subsequently thawed, fixed in formalin, and transferred to 70% ethanol before analyses. Measurements and counts were obtained from x-rays or with digital calipers and rounded to the nearest 0.1 mm (Supplementary Fig. 2). Counts of vertebrae include the Atlas vertebra and the last half centrum (urostyle) being part of the caudal fin skeleton. This method yields higher counts, plus one vertebra, compared to conventional methods. Counts of dorsal and anal fin rays were estimated by counting their supporting pterygiophores from X-rays. The right side outermost gill arch was carefully dissected from each specimen and dyed with Alizarin red[77] to obtain total gill raker counts. The length of the second uppermost gill raker on the lower limb was measured with a reticle eyepiece mounted on a stereomicroscope. PCA on log-transformed measurements was used as an ordination method[78]. PCA was performed using SYSTAT 13. Gill raker length was excluded from PCA due to missing data for several specimens of Slåttersill due to the observed damages (Fig. 2b) and the lack of any obvious difference between samples.

### Organochlorine content and stable isotope analysis

After removing the head and guts, approximately 10 individuals were homogenized to a mixed sample containing muscle tissue, bones, fins, skin, and subcutaneous fat, following the European Union guidelines (EU 2017/644) for food safety analyses. Five and nine pooled samples were prepared using a total of $n = 98$ Slåttersill and $n = 170$ small spring-spawning herring, respectively. 100 g from each pooled sample was transferred to a glass jar, frozen, and transported to ALS Global laboratory in Prague, Czech Republic. After Soxhlet extraction, lipid content and levels of PCDD/Fs and PCB were quantified using high-resolution gas chromatography following US EPA 1613 (PCDD/F) and US EPA 1668 (PCB).

After freeze drying of the homogenized pooled samples, stable isotope analysis was performed with an elemental analyzer−isotope ratio mass spectrometer by the SLU Stable Isotope Laboratory (Swedish University of Agricultural Sciences) with standard procedures and using the same samples as used for measuring organochlorine content.

## Data availability

Sequence data generated in this study have been deposited in the Sequence Read Archive under BioProject PRJNA1137584, while previously published read sets are available under PRJNA642736. Variant data for both SNP-chip and WGS data sets have been deposited in the European Variant Archive under project number PRJEB82679. Frequency data used for heat maps and delta allele frequency contrasts are provided in the Source Data file. Source data are provided with this paper.

## Code availability

All custom code is available in https://github.com/LeifAnderssonLab/Piscivorous_Baltic_Herring (https://doi.org/10.5281/zenodo.14163860).

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

## Acknowledgements

We thank everyone who contributed to the collection and sampling of herring, Lars-Ivan Hållstrand (Hästskär) for informing us about the existence of the Slåttersill population, Christin Appelqvist, Kerstin Johannesson, and the crew at SD511 EROS III for Atlantic herring caught as dead by-catch that we could use for morphological examination, Florian Berg for samples of Atlantic herring from Norway, and Identigen Ltd. for providing extracted genomic DNA. Sincere thanks are also due to Karin Limburg, Debra Driscoll, and Marju Kaljuste for help conducting laboratory analyses, Agnes Karlsson (Stockholm University) for advice on stable isotope analyses and data interpretation, and Anton Larsson for finding the Slåttersill advertisement, Peter and Rosemary Grant, and Nils Ryman for comments on a draft version. The National Genomics Infrastructure (NGI)/Uppsala Genome Center provided service in massive parallel sequencing, and the computational infrastructure was provided by the Swedish National Infrastructure for Computing (SNIC) at UPPMAX, partially funded by the Swedish Research Council (2018-05973). The current project was financially supported by the Baltic Waters Foundation (2110 to L.A. and 2285 to Y.H.), Vetenskapsrådet (2017-02907 to L.A.), Knut and Alice Wallenberg Foundation (KAW 2023.0160 to L.A.), the Swedish Board of Agriculture (3.3.11-04147/2022 to L.A.), Swedish Agency for Marine and Water Management (2702-2023 to Y.H.) and the Swedish Research Council Formas (2023-00297 to U.B.).

## Author contributions

L.A. conceived the study. J.G., M.E.P., and A.W. performed bioinformatic analysis. U.B., J.L., and L.W. collected herring samples and contributed to the interpretation of data. A.C. constructed sequencing libraries. B.D. performed morphological analysis. Y.H. performed otolith analysis. O.M.K. and H.W. were responsible for the analysis of organochlorine content. L.A., J.G., M.E.P. wrote the paper with input from all authors. All authors read and approved the final version.

## Funding

## Competing interests

The authors declare no competing interests.
