## [Transparent Peer Review file · Nature Communications]

Evolution of fast-growing piscivorous herring in the young Baltic Sea

Corresponding Author: Dr Leif Andersson

Version 0:

Reviewer comments:

Reviewer #1

(Remarks to the Author)

I went through the manuscript by Goodall and collaborator entitled “Evolution of fast-growing piscivorous herring in the young Baltic Sea”. This study provides some convincing morphologic and genetic data pointing toward the existence of a piscivorous herring ecotype only found within the Baltic Sea. Specifically, morphological data demonstrate that the piscivorous ecotype differ in size, diet, growth rate, and migration patterns when compared to other herring population present in the Baltic Sea. Population genomics analyses using SNP-chip available for the species show that one population from the piscivorous ecotype is genetically distinct from other herring populations located nearby. Further WGS analyses obtains from 6 additional populations composed by large piscivorous herring show other genomic regions associated to ecotype divergence. Some of these regions are also divergent across piscivorous populations. These differences among populations notably involved a unique haplotype of an already described inversions, with pattern that could correspond to recent recombination event with the other arrangement of the inversions.

Overall, I find the manuscript dense but interesting. However, I think that there are some important mismatches between the results obtains from the SNP-chip and the WGS data analyses that are not well explain in the manuscript. I also feel that the study is very much system oriented. I think the authors could make an effort to broaden the scope of their manuscript to make it suitable for a generalist journal (both in the introduction and data interpretation). Bellow, I give some more information about my feeling:

Major comment:

Link between SNP-chip and WGS analyses:

Two out of the three main genomic regions detected as outlier in Fig 1.e. using the SNP Chip are not clearly outlier in the WGS analyses showed in Extended Data Fig. 5e. However, If I understood correctly, these analyses were made by comparing the same population pair (Slattersill vs Baltic Spring). These different results are concerning because it suggests a lack of reproducibility in the data. The authors need to elaborate a little bit on those mismatches, otherwise, I don't see how these datasets can be accepted for publication.

Comparison between multiple piscivorous populations:

1. The manuscript extensively discusses the genetic differences between seven piscivorous populations of Baltic Sea herring, but I think that more effort could be put to explore and quantify the extent of genetic similarity across piscivorous samples. How much genetic parallelism across piscivorous samples (what is the extent of shared basis of ecotypic divergence) across the Baltic Sea? One could try to detect repeated outlier by doing a multiplication of all the ΔF calculated between piscivorous population and the Baltic Sea spring (similar to what is made for the f_4 statistic), and also by separating the three southern samples vs the other.

2. The PCA in Fig. 3 suggest that the piscivorous populations are spread across two already described lineages (Baltic Sea proper, and Transition Zone). With the analyses conducted here, it is unclear whether the differences presented in figure 3 is specific to piscivorous populations or more generally link to the divergence Atlantic – Transition Zone – Baltic Sea. This question is also true for the recombinant haplotype of the inversion. Is it present in other non-piscivorous population from the Transition Zone?

Broaden the scope of the manuscript:

The introduction and the discussion mostly detail results that are system specific. I think more general paragraph in the introduction on niche expansion and ecotype evolution and what their results add to the already large body of literature on the topic are required to make the manuscript interesting for a general audience not totally familiar with the herring literature.

Minor comments:

First paragraph: I don't find this paragraph very relevant. The first sentence is wrong if we considered that the genetic differentiation can be given by the following formula: $F_{ST} = 1 / (4 N_e m)$, which has no component of R_I . R_I just reduces m . In addition, the end of paragraph, about hybrid speciation, is also somewhat misleading as this study is not about hybrid speciation and rapid evolution. If I was you, I would introduce the concept of ecotype here instead of this paragraph...

Fig. 1c: I am not sure that there is any method linked to this figure. In addition, I find it surprising to fit a linear regression to a growth curve. A reference could be needed here.

Fig. 3c & d, and Fig. 4c.d.e.: I am not sure what are the heatmap representing here. What is the color gradient used within the heatmap linked to? Allele Frequency within each pool with a ref-allele in yellow and a piscivorous-allele color in blue? I find this representation a bit hard to read, mainly because there is a lot of heterogeneity across samples, I would recommend to use some sort of local phylogenetic tree to contrast with the genome-wide average.

Fig. 4. Why using Irish Sea and British samples here instead of non-piscivorous transition zone samples as reference to calculate the F_{ST} ? Would this "piscivorous haplotype" be so different from the one also segregating in the transition zone?

Specific comment:

L47: issue with ref 9

L70: slattersill are instead of is?

L75 if such large herring just represent: remove just?

L80: after colonizing a new environment within the last 8,000 : I would remove this statement as this is not properly inferred. Many lineages / locally adapted allele present in the Baltic Sea can be older than the Baltic Sea itself. Proper demographic inferences are required to understand the evolutionary history of the piscivorous population.

L142 to 165: I found this paragraph/results a bit out of scope and coming out of the blue. This could be easily removed, which will leave room to show more data on the shared genetic basis of piscivorous population.

L203: Mention the number of individuals library here. In addition, I am still not sure why mostly pool-seq analyses were performed here while individuals WGS library were sequenced. Perhaps one sentence is needed somewhere to explain this.

L224: Perhaps add the site number under the bracket for non-Swedish person to understand where these sites are located on the map?

L214 to 220: this could perhaps be a bit expanded to discuss further the reason why the results from the SNP-chip are only partially reproduced in the WGS data (where is the inversion on C17?), and also show more data on the genetic similarity between piscivorous population.

L285: I screened through the manuscript ref 3 by Han et al but could not see such detailed analyses of haplotype diversity within each of the chromosomal rearrangement that could validate that the haplotype of the arrangement presented in Fig. 4 is private to the piscivorous population of the transition zone and is absent from other transition zone population. Could you please show us more details about this?

L750-755: I did not fully understand this. I think this paragraph needs to be reworded to be made clearer. Perhaps start by the purpose of these analyses here, it was difficult to understand what was this about. In addition, I am not sure why doing a log transformation of the data. To me, this is not the data but the residual of the model that should be normally distributed. Finally, by "spawning type" in the factor of the model, do you mean spawning + eco-type?

Reviewer #2

(Remarks to the Author)

This study characterizes a previously undescribed Baltic Sea herring ecotype using a combination of genomic data, otolith analyses, morphological measurements, and chemical/stable isotope analyses, and thus addresses fundamental questions about the evolution of local adaptation and sympatric speciation in a species that not only is almost a model species in evolutionary biology, but also an important cultural and commercial species.

The manuscript was clear and well-written, with impeccable presentation of background, methods and results, and very informative figures. The evidence supporting the authors' interpretation of results was robust and the discussion was concise yet well-developed, and covering and synthesizing all the results in an elegant way. I believe that this paper will be a

reference in the field for future studies looking at the genetic basis of ecotypic differentiation and local adaptation.

Below are two VERY minor comments.

Line 242: Watterson's θ should not be expressed as percentage

Line 248: It is not clear what this measure of divergence refers to as it seems to be derived from F_{st} .

Version 1:

Reviewer comments:

Reviewer #1

(Remarks to the Author)

I was reviewer one in the first round of revision of this manuscript by Goodall et al. and think that the reviewer carefully addressed the concerns I raised in my review. I do feel that the figures are clearer now and provide enough data to validate the inferences made by the manuscript (notably for the recombinant haplotype found in the piscivorous populations from the Transition Zone).

Minor comment:

While I find the literature of the Galapagos radiation very interesting, I don't think the citation about bird Galapagos radiation are super relevant in the context of this manuscript (beyond increasing the self-citations). In addition, I am really not sure that the locally adapted populations of Herring qualifies them as an "evolutionary radiation". There is plenty a very nice example of ecotype evolution in marine species (snails, fish, mammals) that could provide more relevant example (e.g. Killer whale ecotype and diet shift)... Notably, I would like to draw the attention to the recent article on Sea Horses ecotypes from Meyer et al. published in MolEcol <https://hal.science/hal-04306022/document> where they also described recombinant haplotype distinguishing different local population of sea horse, which is a nice comparison to what is found here. (Disclaimer: I am not a co-author on this!).

Diet and dioxine: I understand why this is interesting to find diet specific signature distinguishing piscivorous vs planktivorous populations. However, I still feel that the study of the dioxine level within Baltic Sea Herring could be more properly introduced. Especially if this is an important goal of the manuscript. To me, it not as obvious how dioxine relate to diet as it is for fat content of dN15/DC13.

Response to the reviewers comments

We thank the reviewers for their constructive criticism of our paper that have helped us to improve the manuscript. The most important changes are that we have rewritten the first paragraphs of the Introduction and Discussion as recommended by Reviewer 1. We have also revised some of the figures as described in our point-by-point responses given below in bold.

Reviewer #1 (Remarks to the Author):

I went through the manuscript by Goodall and collaborator entitled “Evolution of fast-growing piscivorous herring in the young Baltic Sea”. This study provides some convincing morphologic and genetic data pointing toward the existence of a piscivorous herring ecotype only found within the Baltic Sea. Specifically, morphological data demonstrate that the piscivorous ecotype differ in size, diet, growth rate, and migration patterns when compared to other herring population present in the Baltic Sea. Population genomics analyses using SNP-chip available for the species show that one population from the piscivorous ecotype is genetically distinct from other herring populations located nearby. Further WGS analyses obtains from 6 additional populations composed by large piscivorous herring show other genomic regions associated to ecotype divergence. Some of these regions are also divergent across piscivorous populations. These differences among populations notably involved a unique haplotype of an already described inversions, with pattern that could correspond to recent recombination event with the other arrangement of the inversions.

Overall, I find the manuscript dense but interesting. However, I think that there are some important mismatches between the results obtains from the SNP-chip and the WGS data analyses that are not well explain in the manuscript. I also feel that the study is very much system oriented. I think the authors could make an effort to broaden the scope of their manuscript to make it suitable for a generalist journal (both in the introduction and data interpretation). Bellow, I give some more information about my feeling:

Major comment:

Link between SNP-chip and WGS analyses:

Two out of the three main genomic regions detected as outlier in Fig 1.e. using the SNP Chip are not clearly outlier in the WGS analyses showed in Extended Data Fig. 5e. However, If I understood correctly, these analyses were made by comparing the same population pair (Slattersill vs Baltic Spring). These different results are concerning because it suggests a lack of reproducibility in the data. The authors need to elaborate a little bit on those mismatches, otherwise, I don't see how these datasets can be accepted for publication.

>>>We assume that the reviewer means Extended Data Fig. 5d (Supplementary Fig. 5d in the revised version). In fact, these are not exactly the same contrasts. In Fig 1e based on SNP-chip data, we compare Slattersill (n=107) with small spring-spawning herring from the Bothnian Sea (n=201), whereas in Supp Fig 5d we compare Slattersill (n=10) with allele frequencies based on pooled sequencing of populations of small spring-spawning Baltic herring from a much broader sampling of the Baltic Sea. (We do not have a sufficient amount of whole genome sequencing data from the Bothnian Sea only, to make this comparison informative.) We think this is the major explanation for the noted difference for instance as regards the signal on Chr17. Furthermore, there is a large difference in sample size which affects how accurate we can estimate delta allele frequencies. We have revised the legend of Supp Fig 5 to make the difference to Fig. 1e clear.

Comparison between multiple piscivorous populations:

1. The manuscript extensively discusses the genetic differences between seven piscivorous populations of Baltic Sea herring, but I think that more effort could be put to explore and quantify the extent of genetic similarity across piscivorous samples. How much genetic parallelism across piscivorous samples (/what is the extent of shared basis of ecotypic divergence) across the Baltic Sea?

One could try to detect repeated outlier by doing a multiplication of all the deltaF calculated between piscivorous population and the Baltic Sea spring (similar to what is made for the f4 statistic), and also by separating the three southern samples vs the other.

>>> In the concluding comments on Line 225-233 we state that we have identified two genetically distinct subpopulations of piscivorous Baltic herring, (i) northern (Slåttersill and large herring from Gävlebukten) and (ii) southern (Stockholm, Östergötland, Kalmar), whereas “Further work is required to determine if the population samples from Blekinge and Finland reflect additional genetically distinct populations of large piscivorous Baltic herring”. The reason for this cautious interpretation of the samples from Bleking and Finland is the small sample sizes we have for these (n=6 and n=7). These are the only samples we have at present from these regions but our ambition is to collect more samples of large herring in the future in order to explore the distribution of piscivorous Baltic herring.

As regards the comparison of the northern and southern piscivorous Baltic herring (BH), we think we have done essentially what the reviewers ask for in the analysis presented in Supp Fig. 6. Here we show that in 50.8% of all 10kb windows across the genome the southern piscivorous BH show a higher sequence identity to the northern piscivorous BH than to any of the two types of planktivorous BH. In contrast, the northern piscivorous show the highest similarity to spring-spawning planktivorous BH, which we think is explained by gene flow from planktivorous BH spawning in the same area. We think this is as far as we can come with existing data.

2. The PCA in Fig. 3 suggest that the piscivorous populations are spread across two already described lineages (Baltic Sea proper, and Transition Zone). With the analyses conducted here, it is unclear whether the differences presented in figure 3 is specific to piscivorous populations or more generally link to the divergence Atlantic – Transition Zone – Baltic Sea. This question is also true for the recombinant haplotype of the inversion. Is it present in other non-piscivorous population from the Transition Zone?

>>>We think the position of the three samples of southern piscivorous Baltic herring is explained by their high frequency of a variant form of the Southern haplotype of the 7.8 Mb chromosome 12 inversion that is more common in the Transition zone than in the Central Baltic Sea. In the revised version of the manuscript we have updated Fig. 4c, d and e and included haplotype information from populations from the Transition zone, both on the Baltic side and the Atlantic side. This analysis shows that at the three diagnostic regions of the Chr12 inversion the haplotypes present in the Transition zone is not similar to the unique variant detected in the southern piscivorous Baltic herring.

Broaden the scope of the manuscript:

The introduction and the discussion mostly detail results that are system specific. I think more general paragraph in the introduction on niche expansion and ecotype evolution and what their results add to the already large body of literature on the topic are require to make the manuscript interesting for a general audience not totally familiar with the herring literature.

>>>This is a good suggestion. We have rewritten the first paragraph of the Introduction and now introduce the concept of adaptative radiations and how they evolve due to niche expansion and character displacement. We agree that this is a better Introduction to the study.

Furthermore, we have written a new first paragraph of the Discussion in an attempt to put our finding in context compared with other species showing niche expansion and reproductive isolation

Minor comments:

First paragraph: I don't find this paragraph very relevant. The first sentence is wrong if we considered that the genetic differentiation can be given by the following formula: $F_{ST} = 1 / (4 N_e \times m)$, which has no component of RI. RI just reduces m. In addition, the end of paragraph, about hybrid speciation, is also somewhat misleading as this study is not about hybrid speciation and rapid evolution. If I was you, I would introduce the concept of ecotype here instead of this paragraph...

>>>We have followed this advice and now introduce the concept of adaptive radiations and

niche expansion in paragraph 1 and we introduce the concept of different ecotypes of Atlantic herring in paragraph 2.

Fig. 1c: I am not sure that there is any method linked to this figure. In addition, I find it surprising to fit a linear regression to a growth curve. A reference could be needed here.

>>>We have decided to remove the linear regression from this figure, the difference in growth curve between these ecotypes is so obvious that no curve fitting is needed.

Fig. 3c & d, and Fig. 4c.d.e.: I am not sure what are the heatmap representing here. What is the color gradient used within the heatmap linked to ? Allele Frequency within each pool with a ref-allele in yellow and a piscivorous-allele color in blue? I find this representation a bit hard to read, mainly because there is lot of heterogeneity across samples, I would recommend to use some sort of local phylogenetic tree to contrast with the genome-wide average.

>>>Fig. 3c&d and Fig 4c-e have been revised and we have corrected the mistake and now includes an explanation for the heatmap. It represents the allele frequency of the reference allele. Furthermore, we have added Neighbor-joining distance trees to Fig. 3 (panel c and e). These clearly support our interpretation that 5 out of 7 large herring population samples share the signal on Chr20 and that the three population samples that we classify as southern piscivorous Baltic herring share a high frequency of a Chr12 inversion haplotype closely related to the one that is common in the waters surrounding Ireland and Great Britain.

Fig4. Why using Irish Sea and British samples here instead of non-piscivorous transition zone samples as reference to calculate the Fst? Would this "piscivorous haplotype" be so different from the one also segregating in the transition zone?

>>>We use the Irish Sea and British samples because the frequency of the Southern haplotype is >90% in this region while it is in the range 25-50% among populations from the Transition zone. However, we have included allele frequencies from the Transition zone in Figs. 4c-e to highlight the uniqueness of the "piscivorous haplotype" for chromosome 12.

Specific comment:

L47: issue with ref 9

>>>We thought this is the way the journal handles reference citations when it is associated with numbers given in superscript. We ask for advise from the Editor if this is correct or not.

L70: slattersill are instead of is?

>>>We prefer "is" here because we refer to a Slåttersill as a type of herring, but we think both are fine. No change.

L75 if such large herring just represent: remove just?

>>>We have followed this minor suggestion.

L80: after colonizing a new environment within the last 8.000 : I would remove this statement as this is not properly infer. Many lineages / locally adapted allele present in the Baltic Sea can be older than the Baltic Sea itself. Proper demographic inferences are required to understand the evolutionary history of the piscivorous population.

>>>We certainly agree that many of the adaptive haplotypes under selection has a much longer evolutionary history than the populations present in the young Baltic Sea. We have shown that in previous studies. However, this statement is valid because both genetic data and the morphological data presented in this study provides strong support for our interpretation that the piscivorous Baltic herring has evolved from planktivorous Baltic herring after the colonization of the Baltic Sea.

L142 to 165: I found this paragraph/results a bit out of scope and coming out of the blue. This could be easily removed, which will leave room to show more data on the shared genetic basis of piscivorous population.

>>>We disagree, we think this is an important part of the study showing that the large Baltic herring must have another diet than the small planktivorous Baltic herring. Furthermore, dioxin contamination is a major concern for the consumption of fish from the Baltic Sea and Baltic herring constitutes about 70% of all fish caught commercially in the Baltic Sea. The great majority of this fish is used for production of fish meal to be used as fish feed in aquaculture. There is in fact a strong societal interest in increasing the human consumption of herring from the Baltic Sea for nutritional and cultural reasons. It is well known that dioxin levels in herring vary between different parts of the Baltic Sea, where the southern basins generally have lower levels. The fact that we now see a variation in dioxin levels within a region opens up a potential opportunity to direct small-scale fishing toward the populations with the lowest levels, minimizing human exposure to dioxins.

L203: Mention the number of individuals library here. In addition, I am still not sure why mostly pool-seq analyses were performed here while individuals WGS library were sequenced. Perhaps one sentence is needed somewhere to explain this.

>>>This information has been added to this sentence:

“Individual libraries for whole genome sequencing were prepared from 103 Baltic herring (Supplementary Table 8)”

We use these data to calculate allele frequencies for comparisons between populations and previously published data based on pool-seq analysis.

L224: Perhaps add the site number under the bracket for non-Swedish person to understand where these sites are located on the map?

>>>We agree this was confusing in the previous version. We have now revised the figures so that we use the same labels as used in the map (Fig. 1a) across all figures and tables.

L214 to 220: this could perhaps be a bit expended to discuss further the reason why the results from the SNP-chip are only partially reproduced in the WGS data (where is the inversion on C17?), and also show more data on the genetic similarity between piscivorous population.

>>>We agree. We have revised the legend of Supplementary Fig. 5 to better explain that the contrast presented here is not the same as the one presented in Fig. 1e (SNP-chip). We also provide information on sample sizes.

L285: I screened through the manuscript ref 3 by Han et al but could not see such detailed analyses of haplotype diversity within each of the chromosomal rearrangement that could validate that the haplotype of the arrangement presented in Fig. 4 is private to the piscivorous population of the transition zone and is absent from other transition zone population. Could you please show us more details about this?

>>>You are absolutely right. We have revised this text and now cite the data presented in Fig. 4c-e in which we have added data for population samples from the Transition zone. This shows that we do not find the unique haplotype at high frequency in spring-spawning Baltic herring, autumn-spawning Baltic herring, northern piscivorous Baltic herring, in Atlantic herring from the Transition zone or in the water surrounding the Irish and British isles.

L750-755: I did not fully understand this. I think this paragraph need to be reword to be made clearer. Perhaps start by the purpose of these analyses here, it was difficult to understand what was this about. In addition, I am not sure why doing a log transformation of the data. To me, this is not the data but the residual of the model that should be normally distributed. Finally, by “spawning type” in the factor of the model, do you mean spawning + eco-type?

>>>>> We have revised this text to clarify the purpose and explained that the response variable otolith Sr:Ca was log transformed to better fit model assumptions and attain symmetrically distributed residuals. We have also replaced “spawning type” with “ecotype” in the model formula to avoid confusion.

Reviewer #2 (Remarks to the Author):

This study characterizes a previously undescribed Baltic Sea herring ecotype using a combination of genomic data, otolith analyses, morphological measurements, and chemical/stable isotope analyses, and thus addresses fundamental questions about the evolution of local adaptation and sympatric speciation in a species that not only is almost a model species in evolutionary biology, but also an important cultural and commercial species.

The manuscript was clear and well-written, with impeccable presentation of background, methods and results, and very informative figures. The evidence supporting the authors' interpretation of results was robust and the discussion was concise yet well-developed, and covering and synthesizing all the results in an elegant way. I believe that this paper will be a reference in the field for future studies looking at the genetic basis of ecotypic differentiation and local adaptation.

Below are two VERY minor comments.

Line 242: Watterson's θ should not be expressed as percentage

>>>**We have corrected this**

Line 248: It is not clear what this measure of divergence refers to as it seems to be derived from F_{ST} .

>>>**It is in fact F_{ST} that is reported here we have changed the sentence as follows:**

“We performed pairwise estimates of genetic differentiation (F_{ST}) among the ecotypes and estimated F_{ST} values to be in the range of 1.6–2.6%,...”

Response to comments from Reviewer 1

We were pleased to learn that Reviewer 1 thinks we have addressed the major issues well. See our comments for the remaining minor issues.

While I find the literature of the Galapagos radiation very interesting, I don't think the citation about bird Galapagos radiation are super relevant in the context of this manuscript (beyond increasing the self-citations). In addition, I am really not sure that the locally adapted populations of Herring qualifies them as an "evolutionary radiation". There is plenty a very nice example of ecotype evolution in marine species (snails, fish, mammals) that could provide more relevant example (e.g. Killer whale ecotype and diet shift)... Notably, I would like to draw the attention to the recent article on Sea Horses ecotypes from Meyer et al. published in MolEcol <https://hal.science/hal-04306022/document> where they also described recombinant haplotype distinguishing different local population of sea horse, which is a nice comparison to what is found here. (Disclaimer: I am not a co-author on this!).
>>>We have shortened the text on Darwin's finches (and reduced self-citations). We prefer to keep a short introduction to the Darwin's finches because they are such a well known example of the evolution of biodiversity involving niche expansion and character displacement, then we continue with salmonids that show a similar evolutionary change after the colonization of freshwater lakes after glaciation. We think this is a relevant introduction to the current study.

We have also added the reference to the paper on seahorses when we discuss gene flux between inversion haplotypes in the Discussion section.

Diet and dioxine: I understand why this is interesting to find diet specific signature distinguishing piscivorous vs planktivorous populations. However, I still feel that the study of the dioxine level within Baltic Sea Herring could be more properly introduced. Especially if this is an important goal of the manuscript. To me, it not as obvious how dioxine relate to diet as it is for fat content of dN15/DC13.

>>>We agree that we should introduce the relevance of the dioxin data for this particular study better. We have added the following first sentence of this paragraph:

"A change in diet may affect the accumulation of pollutants in fish, and we therefore decided to examine this as a test for changed diet in the Slåttersill population."

We have also shortened the text concerning dioxins in the Baltic Sea to keep a focus on the relevance for the characterization of a changed diet in Slåttersill.